# Development of a Nomogram Model for Treatment of Elderly Patients with Locoregionally Advanced Nasopharyngeal Carcinoma

**DOI:** 10.3390/jpm11111065

**Published:** 2021-10-22

**Authors:** Jia Kou, Lu-Lu Zhang, Xing-Li Yang, Dan-Wan Wen, Guan-Qun Zhou, Chen-Fei Wu, Si-Si Xu, Wei-Hong Zheng, Zhen-Yu Qi, Ying Sun, Li Lin

**Affiliations:** 1Department of Radiation Oncology, Sun Yat-sen University Cancer Center, State Key Laboratory of Oncology in South China, Collaborative Innovation Center for Cancer Medicine, Guangdong Key Laboratory of Nasopharyngeal Carcinoma Diagnosis and Therapy, Guangzhou 510060, China; koujia@sysucc.org.cn (J.K.); yangxingl@sysucc.org.cn (X.-L.Y.); wendw@sysucc.org.cn (D.-W.W.); zhougq@sysucc.org.cn (G.-Q.Z.); wucf@sysucc.org.cn (C.-F.W.); xuss@sysucc.org.cn (S.-S.X.); zhengwh@sysucc.org.cn (W.-H.Z.); qizhy@sysucc.org.cn (Z.-Y.Q.); sunying@sysucc.org.cn (Y.S.); 2Department of Molecular Diagnostics, Sun Yat-sen University Cancer Center, State Key Laboratory of Oncology in South China, Collaborative Innovation Center for Cancer Medicine, Guangdong Key Laboratory of Nasopharyngeal Carcinoma Diagnosis and Therapy, Guangzhou 510060, China; zhanglul@sysucc.org.cn

**Keywords:** nasopharyngeal carcinoma, elderly patients, comorbidities, chemotherapy

## Abstract

(**1**) Purpose: This study aims to explore risk-adapted treatment for elderly patients with locoregionally advanced nasopharyngeal carcinoma (LA-NPC) according to their pretreatment risk stratification and the degree of comorbidity. (**2**) Methods: A total of 583 elderly LA-NPC patients diagnosed from January 2011 to January 2018 are retrospectively studied. A nomogram for disease-free survival (DFS) is constructed based on multivariate Cox regression analysis. The performance of the model is evaluated by using the area under the curve (AUC) of the receiver operating characteristic curve and Harrell concordance index (C-index). Then, the entire cohort is divided into different risk groups according to the nomogram cutoff value determined by X-tile analysis. The degree of comorbidities is assessed by the Charlson Comorbidity Index (CCI). Finally, survival rates are estimated and compared by the Kaplan–Meier method and the log-rank test. (**3**) Results: A nomogram for DFS is constructed with T/N classification, Epstein-Barr virus DNA and albumin. The nomogram shows well prognostic performance and significantly outperformed the tumor-node-metastasis staging system for estimating DFS (AUC, 0.710 vs. 0.607; C-index, 0.668 vs. 0.585; both *p* < 0.001). The high-risk group generated by nomogram has significantly poorer survival compared with the low-risk group (3-year DFS, 76.7% vs. 44.6%, *p* < 0.001). For high-risk patients with fewer comorbidities (CCI = 2), chemotherapy combined with radiotherapy is associated with significantly better survival (*p* < 0.05) than radiotherapy alone. (**4**) Conclusion: A prognostic nomogram for DFS is constructed with generating two risk groups. Combining risk stratification and the degree of comorbidities can guide risk-adapted treatment for elderly LA-NPC patients.

## 1. Introduction

Nasopharyngeal carcinoma (NPC) is a special type of head and neck cancer with a high incidence in Southern China and Southeast Asia [1,2]. The age distribution of NPC is unimodal in endemic areas (45–59 years old), while the age distribution is bimodal in non-endemic areas (15–24 years old, 65–79 years old) [3]. With the aging population and the extension of life expectancy, the number of elderly NPC patients will increase [4]. In addition, elderly patients have a higher risk of NPC-related mortality compared with younger patients [5]. Therefore, the management of this cohort needs to be taken seriously.

Currently, treatment decisions for NPC are mainly based on the American Joint Committee on Cancer/Union for International Cancer Control (AJCC/UICC) tumor-node-metastasis (TNM) staging system. National Comprehensive Cancer Network (NCCN) guidelines recommend RT alone for early-stage NPC and radiotherapy combined with platinum-based chemotherapy (CRT) for locoregionally advanced nasopharyngeal carcinoma (LA-NPC). However, the treatment strategies recommended by the international guidelines come from the results of several clinical trials [6,7,8,9] and limited elderly patients are included in these trials due to strict inclusion criteria. Therefore, for elderly patients with LA-NPC, whether combination chemotherapy can further improve survival remains unclear.

Precise risk stratification before treatment is necessary for treatment decisions. At present, the TNM staging system is the most commonly used risk predictor for prognostic prediction and treatment decisions [10]. However, there is prognostic heterogeneity of patients selection based on TNM staging system alone. Therefore, more additional prognostic indicators need to be incorporated to improve the performance of the TNM staging system. In endemic areas, NPC is highly associated with Epstein–Barr virus (EBV) infection. Cell-free (cf) EBV DNA released into the circulatory system by tumor cells can be detected by real-time quantitative polymerase chain reaction (PCR)-based assays [11,12]. Pretreatment cf EBV DNA load, which can reflect tumor burden and is associated with patient survival [13,14,15,16], is now widely used in clinical practice as an ideal prognostic indicator. Moreover, there are other hematological biomarkers (e.g., C-reactive protein (CRP), serum lactate dehydrogenase (LDH), hemoglobin (HGB) and albumin (ALB)) that have been demonstrated to have clinical utility for prognostication [17,18,19,20,21].

Comorbidities are also the important factors when developing treatment strategies, especially for the elderly who are more likely to suffer from co-existing ailments and decreasing organ function. Patients with severe comorbidities are unable to tolerate intensive treatment and are vulnerable to treatment-related complications and death from noncancerous diseases [17,22,23]. All of these issues affect the formulation of treatment strategies and the final survival outcome of patients. 

Against this background, in this study, a pretreatment prognostic nomogram is constructed for elderly LA-NPC patients by combining the TNM staging system, cf EBV DNA and other clinical factors. This comprehensive prognostic model can effectively predict DFS in elderly patients with LA-NPC. Furthermore, risk-adapted treatment schemes are explored according to risk classification determined by the nomogram and the complications of patients.

## 2. Materials and Methods

### 2.1. Data Extraction and Patient Selection

Since 2015, Sun Yat-sen University Cancer Center (SYSUCC) has established an automatic and dynamic big data intelligence platform that updates and integrates detailed electronic health record data extracted from daily healthcare systems. This allows oncologists to select eligible patients accurately, extract examination and therapeutic information and track the follow-up of patients. Using this platform, a total of 583 elderly patients with histologically proven and non-disseminated NPC who are newly diagnosed from January 2011 to January 2018 are reviewed. The inclusion criteria are: (a) age 65 years or older; (b) LA-NPC (stages III-IVA) as determined by 8th edition of the AJCC/UICC staging system; (c) received radical radiotherapy combined with or without chemotherapy (CRT/RT); (d) complete clinicopathologic and treatment data. 

Smoking and drinking habits are extracted from medical records based on patients’ self-reports. Medical records include whether or not to smoke/drink, daily amount of smoking/alcohol consumption, duration of smoking/drinking, whether or not to quit smoking/drinking and duration of smoking/drinking cessation. Patients are considered as smokers or drinkers as long as they ever smoked or drank alcohol. This study is approved by the institutional review board of SYSUCC (approval number: B2020-267) and informed consent is waived as this retrospective study is based on an analysis of patients’ routine clinical and treatment data. 

### 2.2. Examinations and Treatment Protocols

All patients undergo baseline examinations before initiating treatment. The severity of comorbidity at diagnosis is determined by the Charlson Comorbidity Index (CCI), which is first introduced in 1987 and contains nineteen medical conditions and age factors with a weighted score based on the adjusted risk of one-year mortality [24]. The sum of all the weighted scores gives a single comorbidity score for each patient (the evaluation method is described in detail on the website: https://www.mdcalc.com/charlson-comorbidity-index-cci (accessed on 20 September 2021)). A score of zero indicates no comorbidities are found. The higher the score, the more likely the predicted outcome will result in higher resource utilization or mortality. For every decade, 1 point is added to risk if the patient is over 40 years old and the “age points” are added to the CCI score. In this study, no patient scores less than 2.

In this study, all patients receive radical radiotherapy. The prescribed doses range from 60 to 70 Gy with the daily fraction dose ranging from 2.00 to 2.43 Gy, five times a week for 6–7 weeks. Chemotherapy is not administered in some patients due to the rejection of patients or contraindication from medical comorbidities. Details of the examinations and chemotherapy protocols are described in the Appendix A.

### 2.3. Follow-Up and Endpoints

Patients attend follow-up appointments at least every 3 months for the first 3 years after the end of treatment and then every 6 months thereafter until death. Follow-up duration is calculated from the day of treatment initiation to the last day of contact or death. The primary endpoint is disease-free survival (DFS, defined as the time from treatment initiation to the first failure or death from any cause). The secondary endpoint is overall survival (OS, defined as the time from the date of treatment initiation to death from any cause). 

### 2.4. Statistical Analysis

The CCI is divided into categorical variables based on a cutoff value determined by receiver operating characteristic (ROC) analysis. Other continuous variables are converted into categorical variables according to clinical cutoff points (ALB, LDH, HGB and CRP) or findings reported in previous studies (cf EBV DNA load) [25,26]. Cumulative survival rates are estimated by using the Kaplan–Meier method and compared by using the log-rank tests. Univariate and multivariate COX regression analysis is performed to define DFS predictors, then a forest plot is generated to demonstrate the results of the multivariate analysis.

Based on the significant prognostic factors from multivariate Cox regression analysis, the nomogram for DFS is then developed. The calibration curves, which indicate the calibration ability of the nomogram, are assessed graphically by plotting the actual observed survival rates against the nomogram-predicted survival rates via a bootstrap method with 1000 resamples. The discrimination performance of the nomogram is measured quantitatively by Harrell’s concordance indices (C-index), which are measured by using R software (version 3.4.3.) and the Hmisc package. Moreover, the prognostic performance of the nomogram model, significant prognostic factors from the multivariate analysis and the 8th AJCC/UICC TNM staging system are compared using C-index and the area under the ROC curve (AUC).

The X-tile software are used (version 3.6.1; Yale University School of Medicine, New Haven, CT, USA) to determine the optimal cutoff value of the nomogram and to divide the entire cohort into high-risk and low-risk groups. The Pearson χ2 test or Fisher’s exact test is applied to assess the basic characteristics of these two groups. In order to develop the risk-adapted treatment regimen for elderly LA-NPC patients, a subgroup analysis according to the severity of complications in the two groups are performed to compare RT and CRT. All statistical analyses are performed by using SPSS, version 22.0 (SPSS Inc., Chicago, IL, USA) or the rms package in RStudio version, 1.3.1093, unless otherwise specified. Statistical significance is set at a two-tailed *p* < 0.05.

## 3. Results

### 3.1. Patient Characteristics and Follow-Up

A total of 583 eligible patients are included in this retrospective analysis. The baseline characteristics of these patients are shown in Table 1. The median age at diagnosis is 68 years old (65–91 years old). Of all the patients, 43.2% (252/583) have a CCI score of 2 and 56.8% (331/583) have a CCI score above 2. The median follow-up duration of the entire cohort is 47.1 months (2.7–149.5 months). During the follow-up period, 61 (10.5%) and 104 (17.8%) patients develop locoregional relapses and distant metastases, respectively; and 196 (33.6%) patients die from any cause. The detailed causes of death are described in the Appendix A. The 3-year and 5-year DFS for the entire cohort are 69.8% and 60.3%, respectively. The 3-year and 5-year OS for the entire cohort are 79.1% and 65.8%, respectively.

### 3.2. Development and Validation of Nomograms for DFS

In univariate analysis, T classification, N classification, TNM overall stage, pretreatment cf EBV DNA load, ALB, CRP and HGB are significant prognosticators for DFS (Table 1). As TNM overall stage is a combination of T and N classification, we include only T/N classification, pretreatment cf EBV DNA, ALB, CRP and HGB in the stepwise multivariate analysis. T classification, N classification, pretreatment cf EBV DNA load and ALB remain the significant and independent prognosticators in multivariate analysis (Figure 1). A prognostic nomogram model for DFS is constructed based on these four predictors factors (Figure 2a). The point assignment is described in detail in Appendix A. Calibration plots present an excellent agreement between the nomogram predicted DFS and the actual observed DFS (Figure 2b,c). The C-index of the nomogram (0.668; 95% CI, 0.633–0.703) is significantly better than cf EBV DNA load (0.619; 95% CI, 0.587–0.652; *p* < 0.001), the 8th edition of the TNM staging system (0.585; 95% CI, 0.552–0.618; *p* < 0.001), T classification (0.561; 95% CI, 0.528–0.595; *p* < 0.001), N classification (0.584; 95% CI, 0.548–0.620; *p* < 0.001) and ALB (0.549; 95% CI, 0.520–0.578; *p* < 0.001) in predicting DFS (Table 2).

The ROC analysis proves that the nomogram is superior to the other clinical factors in predicting DFS (Table 2 and Appendix A). The AUC of the nomogram (0.710; 95% CI, 0.671–0.746) is higher than that of EBV DNA (0.656; 95% CI, 0.617–0.695; *p* < 0.001), the 8th edition of TNM staging system (0.607; 95% CI, 0.567–0.647; *p* < 0.001), T classification (0.588; 95% CI, 0.547–0.628; *p* < 0.001), N classification (0.590; 95% CI, 0.549–0.630; *p* < 0.001) and ALB (0.558; 95% CI, 0.517–0.599; *p* < 0.001).

### 3.3. Risk Stratification Based on the Nomogram

According to the results of X-tile software (Appendix A), a nomogram score of 114 is the optimal cutoff value for predicting DFS and the entire cohort is divided into high-risk (*n* = 283) and low-risk groups (*n* = 300). The clinical characteristics of patients in both groups are detailed in Appendix A. The patients in the high-risk group present poorer survival outcomes compared with those in the low-risk group at all survival endpoints. The 3-year DFS of the low-risk group and the high-risk group are 82.7% and 56.9% and the 5-year DFS of the low-risk group and the high-risk group are 76.7% and 45.4% (*p* < 0.001, Figure 3a). The 3-year OS of the low-risk group and the high-risk group are 89.5% and 69.2% and the 5-year OS of the low-risk group and the high-risk group are 81.5% and 51.6% (*p* < 0.001, Figure 3a).

### 3.4. Risk-Adapted Treatment Strategies Based on the Nomogram and CCI

Firstly, the efficacy of RT and CRT in two risk groups generated by nomogram model are compared. In low-risk group, patients who receive CRT do not show significantly better DFS and OS compared with those who receive RT (CRT vs. RT, 3-year DFS: 84.3% vs. 80.8%, *p* = 0.338; 3-year OS: 91.4% vs. 87.2%, *p* = 0.276; Figure 3b). In high-risk group, CRT are significantly superior to RT alone in terms of DFS and OS (CRT vs. RT, 3-year DFS: 66.2% vs. 43.2%, *p* = 0.001; 3-year OS: 75.1% vs. 59.4%, *p* = 0.009; Figure 3c).

In addition, considering the influence of patient comorbidities on the selection of treatment intensity, subgroup analysis is performed, which is based on the severity of complications in the low- and high-risk groups to explore the optimal treatment regimen. CCI score is used to describe the severity of complications and converted into a categorical variable according to the score of 2, which is the optimal cutoff value for DFS prediction determined by ROC analysis. In the low-risk group with CCI = 2, no significant differences in survival outcome are observed among the CRT and RT (CRT vs. RT, 3-year DFS: 80.5% vs. 82.3%, *p* = 0.855; 3-year OS: 89.2% vs. 85.6%, *p* = 0.895; Figure 4a). Similarly, in the low-risk group with CCI >2, no significant differences in survival outcome are observed between the two treatment regimens (CRT vs. RT, 3-year DFS: 86.7% vs. 80.1%, *p* = 0.149; 3-year OS: 92.0% vs. 87.8%, *p* = 0.169; Figure 4b).

In the high-risk group, the patients with CCI = 2 who receive CRT are significantly superior to those who receive RT in terms of DFS and OS (CRT vs. RT, 3-year DFS: 71.9% vs. 40.6%, *p* = 0.006; 3-year OS: CRT vs. RT, 79.4% vs. 56.3%, *p* = 0.029; Figure 4c). For the patients with CCI >2, there are no significant differences between the CRT and RT in term of DFS and OS (CRT vs. RT, 3-year DFS: 57.7% vs. 42.9%, *p* = 0.132; 3-year OS: 69.7% vs. 59.4%, *p* = 0.319; Figure 4d).

## 4. Discussion

Elderly patients with LA-NPC have an unsatisfactory survival outcome and current treatment guidelines have mostly been designed based on clinical trials with limited elderly patients included [6,7,8,27]. Whether elderly LA-NPC patients will benefit from treatment strategies recommended by guidelines remains unclear. In this study, a prognostic nomogram in elderly LA-NPC patients is constructed by incorporating the T classification, N classification, pretreatment cf EBV DNA load and other clinical prognostic factors. In addition, risk-adapted treatment strategies for elderly LA-NPC patients are further explored based on the risk-stratification generated by the nomogram and the degree of comorbidities. This prognostic nomogram model can improve prognostic efficacy compared with the traditional staging system and guide the individualized risk-adapted treatment for elderly LA-NPC patients. To our knowledge, this is the first study to propose the risk-adapted treatment for elderly LA-NPC patients by combining comorbidities of patients with pre-treatment prognostic model.

At present, cumulative retrospective studies have explored treatment strategies for elderly patients with LA-NPC. In the era of conventional two-dimensional radiotherapy (2D-RT), Liu and Zeng et al. reported that chemotherapy combined with radiotherapy improves survival for elderly LA-NPC patients compared with radiotherapy alone [28,29]. However, with the development of radiotherapy technology and the wide application of intensity-modulated radiotherapy (IMRT) in clinical practice, several retrospective studies have shown no benefit on survival with the addition of chemotherapy in elderly LA-NPC patients receiving IMRT [30,31,32]. Notably, these retrospective studies are conducted in whole cohorts of elderly LA-NPC patients, thereby ignoring the clinical heterogeneity of patients, which is an important factor influencing treatment decisions. Due to this heterogeneity, patients with the same TNM stage, who receive the same treatment regimen have markedly different survival outcomes, which highlights the importance of a prognostic model that integrates both clinical and molecular risk in identifying truly high-risk patients for intensive treatment.

Therefore, in this study, the TNM staging system, pre-treatment cf EBV DNA load and various other clinical factors are comprehensively considered in order to create a risk classification to develop risk-adapted treatment strategies for elderly LA-NPC patients. Multivariate Cox regression analysis demonstrates that T classification, N classification, pretreatment cf EBV DNA load and ALB are independently associated with DFS. The TNM staging system, which well reflects tumor burden, is widely recognized and the most commonly used pre-treatment risk stratification system. In addition, pretreatment cf EBV DNA load is a biomarker that can reflect tumor burden and patient prognosis and it is considered as the complement to the TNM staging system [13]. A previous study by Lv et al. supported the concept of utilizing cf EBV DNA for individualization of chemotherapy intensity in patients with LA-NPC [33]. In addition, the results of this study suggest that lower ALB levels are associated with poorer survival in elderly LA-NPC patients. Low ALB levels, to some extent, reflect malnutrition and have been associated with poor outcomes in cancer patients [23,34].

Next, a prognostic nomogram is developed by integrating T classification, N classification, pretreatment cf EBV DNA load and ALB. The high-risk and low-risk groups generated by the nomogram have significantly different survival outcomes. Compared with the TNM staging system and the above significant risk predictors, the prognostic nomogram model has better prognostic efficacy and can screen out high-risk patients who will benefit from intensive treatment.

It is well-known that elderly patients are associated with declining physiological function and increasing comorbidity rate, which can alter the pharmacokinetics of many commonly used chemotherapeutic agents. Then, the sensitivity of elderly patients to radiotherapy and chemotherapy will decrease [35,36,37,38]. The value of comorbidities in guiding treatment decisions for patients with NPC has been determined in recent years [28,39]. Based on the risk stratification generated by the prognostic nomogram, the influence of comorbidity on treatment decisions is further considered. In this study, CCI scores are used to describe the degree of patient’s complications [24]. The results show that only patients in the high-risk group with CCI = 2 can benefit from the additional chemotherapy. However, the results do not show improved benefit on survival from additional chemotherapy for high-risk patients with CCI >2 and low-risk patients. A study by Liu et al. also reported that additional chemotherapy significantly improves the 5-year OS in elderly patients with LA-NPC, but not in those with more complications [28]. This is because severe comorbidities may increase the toxicity of specific treatments and shorten remaining life expectancy gained from therapy.

However, this study has several limitations. Firstly, the treatment protocols of patients have a certain heterogeneity due to physician biases, which are unavoidable in retrospective studies. To best address this, patients who receive a non-standard treatment regimen are excluded, At the same time, the treatment intensity and regimens consistent are also tried to keep. Of note, the advantage of retrospective studies is that they can reflect the treatments outcomes in real-world scenarios. Secondly, no data on treatment-related toxicity is reported in this study. Although the results show only 11% of elderly patients die from therapeutic toxicity, treatment-related toxicity is not negligible in determining chemotherapy due to the fact that elderly patients may value the quality of life over the extension of life. To sum up, the results in this study should be treated cautiously and verified in a well-designed prospective clinical study.

## 5. Conclusions

In conclusion, a prognostic nomogram for DFS in elderly patients with LA-NPC is constructed by incorporating T classification, N classification, cf EBV DNA load and ALB. By combining risk stratification generated by the nomogram and degree of comorbidities, individual risk-adaption treatment for elderly LA-NPC patients can be tailored. High-risk patients with CCI = 2 may benefit from additional chemotherapy, while high-risk patients with CCI > 2 and low-risk patients are unlikely to benefit from an intensive treatment regimen.

## Figures and Tables

**Figure 1 jpm-11-01065-f001:**
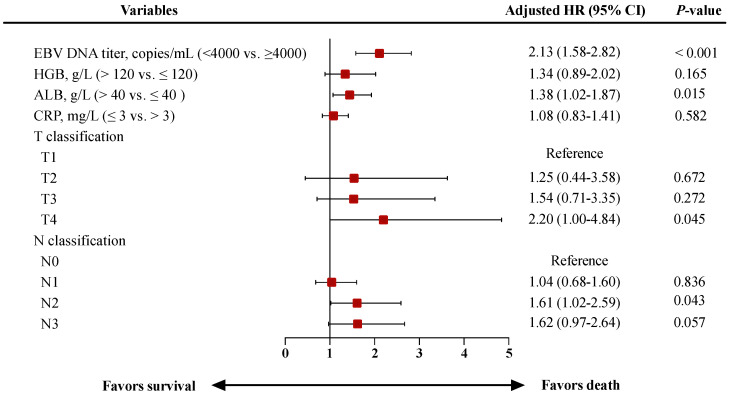
Forest plot shows the results of the multivariate analysis for DFS in elderly patients with LA-NPC. Red squares indicate HRs and horizontal bars indicate 95% CIs. LA-NPC, locoregionally advanced nasopharyngeal carcinoma; EBV, Epstein–Barr virus; DNA, deoxyribonucleic acid; HGB, hemoglobin; ALB, albumin; CRP, C-reactive protein; HR, hazard ratio; CI, confidence interval. DFS, disease-free survival.

**Figure 2 jpm-11-01065-f002:**
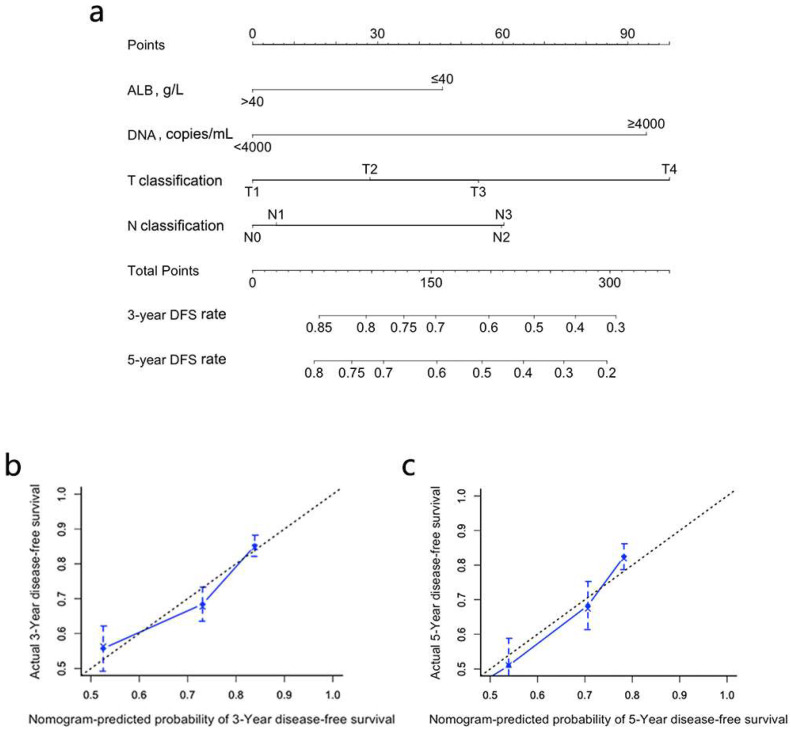
Prognostic nomogram (**a**) and calibration plots (**b,c**) for 3-year and 5-year DFS in elderly patients with LA-NPC. The *X*-axis represents the probability of 3-year DFS predicted by the nomogram and the *Y*-axis represents the actual observed probability of 3-year DFS. The dotted line indicates that the predicted and observed survival probability is consistent. NPC, nasopharyngeal carcinoma; EBV, Epstein–Barr virus; ALB, albumin; DFS, disease-free survival; OS, overall survival.

**Figure 3 jpm-11-01065-f003:**
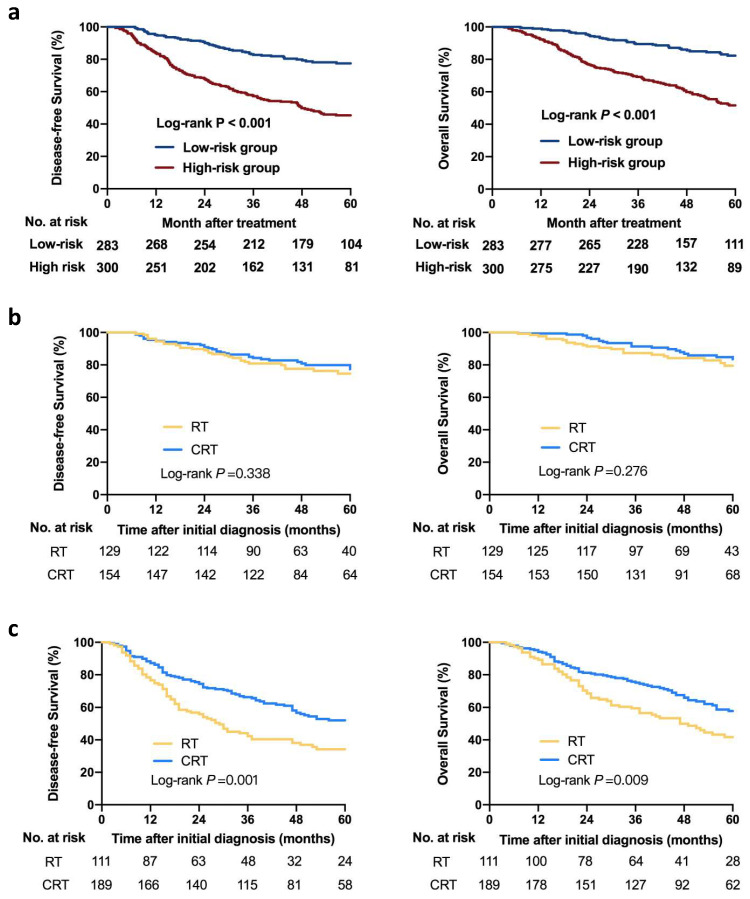
Survival outcomes of the high-risk and low-risk groups generated by the prognostic nomogram. Kaplan–Meier curves for DFS and OS for the two groups (**a**). Survival outcomes of different treatment regimens in the low-risk group (**b**) and the high-risk group (**c**). RT, radiotherapy; CRT, radiotherapy combined with chemotherapy; DFS, disease-free survival; OS, overall survival.

**Figure 4 jpm-11-01065-f004:**
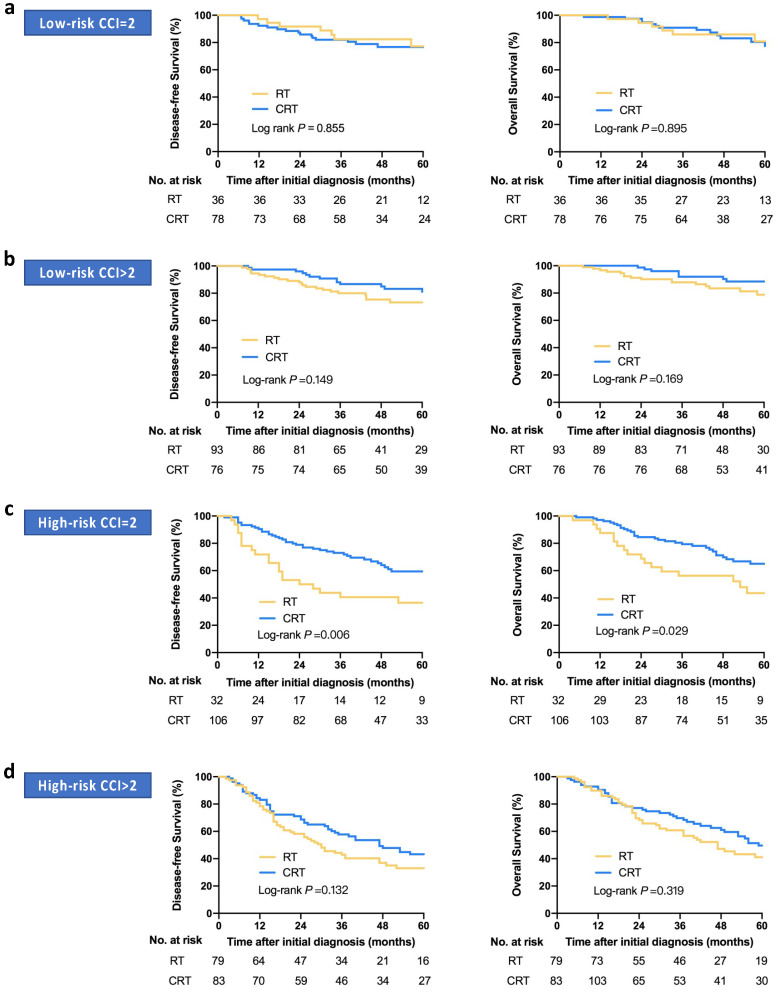
Kaplan–Meier DFS and OS curves of different treatment regimens in subgroups analysis. Survival outcomes of different treatment regimens in the low-risk groups with CCI = 2 (**a**), in the low-risk groups with CCI >2 (**b**), in the high-risk groups with CCI = 2 (**c**) and in the high-risk groups with CCI >2 (**d**); RT, radiotherapy; CRT, radiotherapy combined with chemotherapy; DFS, disease-free survival; OS, overall survival.

**Table 1 jpm-11-01065-t001:** Clinicopathologic characteristics and univariate analysis for DFS in the 583 elderly patients with locoregionally advanced nasopharyngeal carcinoma.

Characteristics	Entire Cohort No. (%)	*p*-Value	HR (95% CI)
Age, years			
≤68	336 (57.6%)		Ref.
>68	247 (42.4%)	0.076	1.26 (0.97–1.63)
Gender			
Male	463 (79.4)		Ref.
Female	120 (20.6)	0.373	0.86 (0.61–1.20)
Histological type (WHO) ^1^			
Type I–II	18 (3.1)		Ref.
Type III	565 (96.9)	0.749	1.14 (0.51–2.57)
Smoking			
Yes	232 (39.8)		Ref.
No	351 (60.2)	0.992	1.00 (0.77–1.30)
Drinking			
Yes	92 (15.8)		Ref.
No	491 (84.2)	0.970	0.99 (0.70–1.42)
Family history of NPC			
Yes	115 (19.7)		Ref.
No	468 (80.3)	0.321	0.84 (0.60–1.18)
EBV DNA, copies/mL ^2^			
<4000	320 (54.9)		Ref.
≥4000	263 (45.1)	< 0.001	2.65 (2.03–3.47)
LDH, IU/L ^2^			
≤250	540 (92.6)		Ref.
>250	43 (7.4)	0.243	1.32 (0.83–2.12)
HGB, g/L ^2^			
≤120	51 (8.7)		Ref.
>120	532 (91.3)	0.024	0.63 (0.42–0.94)
CRP, mg/L ^2^			
≤3	349 (59.9)		Ref.
>3	234 (40.1)	0.043	1.31 (1.01–1.69)
ALB, g/L ^2^			
≤40	117 (20.1)		Ref.
>40	466 (79.9)	< 0.001	0.59 (0.44–0.78)
T classification ^3^			
T4	161 (27.5)		Ref.
T1	19 (3.3)	0.146	0.56 (0.26–1.22)
T2	18 (3.1)	0.369	0.70 (0.32–1.51)
T3	385 (66.0)	< 0.001	0.53 (0.41–0.69)
N classification ^3^			
N0	92 (15.8)		Ref.
N1	265 (45.5)	0.362	1.22 (0.80–1.87)
N2	138 (23.7)	0.003	1.97 (1.26–3.08)
N3	88 (15.1)	< 0.001	2.34 (1.46–3.77)
Overall stage ^3^			
III	358 (61.4)		Ref.
IVa	225 (38.6)	< 0.001	1.99 (1.54–2.58)
Radiotherapy techniques			
IMRT	569 (97.6%)		Ref.
2D-RT/3D-CRT	14 (2.4%)	0.752	0.88 (0.38–1.97)
CCI ^4^			
=2	252 (43.2)	-	-
>2	331 (56.8)	-	-
Treatment modality			
RT alone	240 (41.2)	-	-
CRT	343 (58.8)	-	-

Abbreviations: No., number; DFS, disease-free survival; HR, hazard ratio; CI, confidence interval; Ref., reference; WHO, World Health Organization; CCI, Charlson Comorbidity Index; EBV, Epstein-Barr virus; LDH, serum lactate dehydrogenase level; HGB, hemoglobin; CRP, C-reactive protein; ALB, albumin; RT, radiotherapy; CRT, radiotherapy combined with chemotherapy; 2D-RT, two-dimensional radiotherapy; 3D-CRT, three-dimensional conformal radiotherapy; IMRT, intensity-modulated radiotherapy. ^1^ WHO Type I, keratinizing, WHO Type II, non-keratinizing (differentiated), WHO Type III, non-keratinizing (undifferentiated). ^2^ All variables are measured before treatment. ^3^ According to the 8th edition of the AJCC/UICC staging system. ^4^ Each decade of age over 40 add 1 point to risk and the “age points” are added to the score from the Charlson Comorbidity Index. Therefore, no patient in our study had a CCI score below 2.

**Table 2 jpm-11-01065-t002:** C-index and AUC of prognostic model and single risk factors for predicting DFS in elderly patients with LA-NPC.

Risk Factors	C-Index (95% CI)	*p*-Value	AUC (95% CI)	*p*-Value
Prognostic models				
Nomogram	0.668 (0.633–0.703)	Ref.	0.710 (0.671–0.746)	Ref.
8th TNM staging system ^1^	0.585 (0.552–0.618)	<0.001	0.607 (0.567–0.647)	<0.001
Single risk factors				
EBV DNA	0.619 (0.587–0.652)	<0.001	0.656 (0.617–0.695)	<0.001
T classification ^1^	0.561 (0.528–0.595)	<0.001	0.588 (0.547–0.628)	<0.001
N classification ^1^	0.584 (0.548–0.620)	<0.001	0.590 (0.549–0.630)	<0.001
ALB	0.549 (0.520–0.578)	<0.001	0.558 (0.517–0.599)	<0.001

Abbreviations: DFS, disease-free survival; C-index, Harrell’s concordance indices; AUC, area under the receiver operator characteristic curve; CI, confidence intervals; Ref., reference; EBV, Epstein-Barr virus; ALB, albumin; LA-NPC, locoregionally advanced nasopharyngeal carcinoma; ^1^ According to the 8th edition of the AJCC/UICC staging system.

## Data Availability

The datasets during the current study have been deposited in the Research Data Deposit public platform (www.researchdata.org.cn (accessed on 20 September 2021)) and are available from the corresponding author on reasonable request.

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
