# Peer review of "Development of a Nomogram Model for Treatment of Elderly Patients with Locoregionally Advanced Nasopharyngeal Carcinoma"

_jpm, 2021, doi:10.3390/jpm11111065_

Round 1
Reviewer 1 Report
The present paper deals with elderly NPC patient selection for curative treatment. The Authors built a predictive nomogram for DFS based on T/N classification, EBV DNA load and albumin. The paper is well written and the research topic is of relevant interest for the field. I only suggest language revision with harmonization of tenses.
Author Response
Thank you very much for your suggestion. We have double-checked the entire manuscript carefully and corrected the tenses and grammar errors.
Reviewer 2 Report
This article is well described about the outcome of treatment for locoregionally advanced Nphx ca. in elderly patient. I'm very interested about this article and also agree that this results are considered to be valuable.
However, I have some questions about your article.
- Did all study participants conducted same RTx method? (ex, IMRT, 3DCRT) If not, I think this might seriously affects the results of this study.
- Your authors emphasize that traditional staging system did not reflect intra-tumoral heterogeneity, so it did not fully predict the prognosis. However, I think your mentioned factors are not associated with intra-tumoral heterogeneity. It is just some clinical characteristics. What do you think about it?
- I have a minor question that how about the definition about smoking and drinking?
Author Response
1、Did all study participants conducted same RTx method? (ex, IMRT, 3DCRT) If not, I think this might seriously affects the results of this study.
Response: Thank you for your kind reminder. It is true that not all patients in our study received IMRT, which is described in detail in the supplementary materials: “Of the total patients, 97.6% (569/583) of patients received IMRT, and only 13 (2.2%) and 1 (0.2%) patients received conventional 2D-RT and 3D conformal RT (3D-CRT).” We believe that although there is a slight difference in radiotherapy technology, it is difficult to have an impact on the results of this study, mainly because the proportion of patients receiving 2D-RT/3D-CRT in this study is very low (only 2.4%). In this regard, we have included the radiotherapy technique in the univariate COX regression analysis, and the results indicated that there was no significant difference in DFS between IMRT and 2D-RT/3D-CRT (HR 0.877, 95%CI 0.388-1.979, P=0.752). We have also added this result to the revised manuscript (Page 5, Table 1).
2、Your authors emphasize that traditional staging system did not reflect intra-tumoral heterogeneity, so it did not fully predict the prognosis. However, I think your mentioned factors are not associated with intra-tumoral heterogeneity. It is just some clinical characteristics. What do you think about it?
Response: We thank the reviewer for pointing out our inaccurate statement that circulating cell-free Epstein–Barr virus DNA (cf EBV DNA) could reflect intra-tumoral heterogeneity. Indeed, none of the prognostic factors we include in this study reflect intra-tumor heterogeneity. However, it is generally believed that cf EBV DNA, is a fragment of DNA released into the circulatory system by NPC cells, can reflect tumor burden and is associated with patient survival [1-2]. Therefore, cf EBV DNA has an important role in patient prognostication and treatment response monitoring. We have checked the entire manuscript and revised all similar errors. Your suggestion will make our revised manuscript more rigorous(Page 2, Line 60-63 and Page 10-11, Line 302, 310-312, 323-328, 336).
References:
[1] Lo YM, Chan AT, Chan LY, et al. Molecular prognostication of nasopharyngeal carcinoma by quantitative analysis of circulating Epstein-Barr virus DNA. Cancer Res. 2000;60:6878-6881.
[2] Ma BB, King A, Lo YM, et al. Relationship between pretreatment level of plasma Epstein-Barr virus DNA, tumor burden, and meta- bolic activity in advanced nasopharyngeal carcinoma. Int J Radiat Oncol Biol Phys. 2006;66:714-720.
3、I have a minor question that how about the definition about smoking and drinking?
Response: Thank you very much for your question. Smoking and drinking habits are extracted from medical records based on patients' self-reports. Medical records include whether or not to smoke/drink, daily amount of smoking/alcohol consumption, duration of smoking/drinking, whether or not to quit smoking/drinking, and duration of smoking/drinking cessation. Patients are considered as smokers or drinkers as long as they ever smoked or drank alcohol. We have added this detailed description to the methods section of the revised manuscript (Page 2-3, Line 97-110).